# Optimization of EPA-Nattokinase Nanoemulsions Processed by High-Pressure Homogenization to Enhance Stability and Thrombolytic Efficacy

**DOI:** 10.3390/foods14203482

**Published:** 2025-10-12

**Authors:** Jiaxing Wang, Shanshan Xu, Liang Chen, Pingan Zheng, Ru Song, Yan Song, Jipeng Sun, Bin Zhang

**Affiliations:** 1School of Food Science and Pharmacy, Zhejiang Ocean University, Zhoushan 316022, China; 18868005756@163.com (J.W.); xushanshan0415@zjou.edu.cn (S.X.); chengliang@zjou.edu.cn (L.C.); rusong@zjou.edu.cn (R.S.); songy@zjou.edu.cn (Y.S.); 2Research Office of Marine Biological Resources Utilization and Development, Zhejiang Marine Development Research Institute, Zhoushan 316021, China; zheng_ping_an@126.com

**Keywords:** nanoemulsion, EPA, synergistic delivery, thrombosis prevention

## Abstract

This study leverages nanoemulsion technology to engineer a novel liquid formulation combining Eicosapentaenoic acid (EPA) and Nattokinase (NK), aiming to enhance their application potential in functional foods. Both EPA and NK are well recognized for their pronounced anti-thrombotic, anti-inflammatory, and lipid-lowering properties, which are critical for the prevention and management of cardiovascular diseases. However, their practical application in functional foods is hampered by inadequate gastrointestinal stability and suboptimal bioavailability. Here, an EPA-NK nanoemulsion was fabricated using high-pressure homogenization technology. We systematically evaluated its environmental stability, anti-thrombotic activity, and intervention efficacy against carrageenan-induced black-tail thrombosis. The results demonstrated that the nanoemulsion not only enhanced the potential for oral bioavailability based on in vitro stability and preliminary in vivo efficacy trends of EPA and NK but also notably potentiated their synergistic anti-thrombotic efficacy, thereby providing robust theoretical and technical support for the development of next-generation health-promoting functional foods targeting thrombotic disorders.

## 1. Introduction

Eicosapentaenoic acid (EPA), an n-3 long-chain polyunsaturated fatty acid abundant in cold-water fish, algae and some plant oils, is essential for cardiovascular and neurological health [1]. Clinical-grade EPA-EE (≥96% EPA) lowers very-low-density lipoprotein triglyceride synthesis, reducing cardiovascular risk [2]. Beyond cardioprotection, EPA improves obesity [3], exhibits anti-tumour activity by suppressing angiogenesis [4,5], and enhances antioxidant capacity [6]. It modulates inflammation [7], alleviates neuropathic pain [8] and protects renal function [9]. However, EPA’s high degree of unsaturation renders it prone to oxidation when exposed to light, heat or air, and its lipophilicity limits bioavailability [10].

NK, a 27.7 kDa serine protease from Bacillus subtilis natto, directly degrades fibrin without affecting circulating fibrinogen, offering superior safety and efficacy over streptokinase or urokinase [11,12]. NK decreases plasminogen activator inhibitor-1 (PAI-1) and increases tissue plasminogen activator (tPA), enhancing endogenous fibrinolysis [13]; it also elevates antithrombin levels and inhibits collagen- or thrombin-induced platelet aggregation, thereby preventing thrombosis [14]. Additional benefits include ACE inhibition and blood-pressure reduction [15], anti-lipidaemic activity [16] and neuroprotection [17]. Yet, low oral bioavailability and limited gastrointestinal stability restrict NK’s clinical utility; microencapsulation [18] and directed mutagenesis [19] are being explored to overcome these hurdles.

Nanoemulsions—kinetically stable colloidal systems with 10–500 nm droplets—can solubilise lipophilic actives and protect labile compounds, improving their stability, bioavailability and controlled release [20,21,22,23,24,25].

The present study employs EPA as the oil phase to harness the synergistic thrombolytic potential of EPA and NK while mitigating their inherent instability. High-pressure homogenisation was used to generate an EPA-NK nanoemulsion with high enzymatic activity retention and colloidal stability. The formulation was systematically optimised, its performance under environmental stresses (temperature, pH, ionic strength) and accelerated oxidation was evaluated, and excipients enhancing stability were screened. The optimised nanoemulsion was finally assessed for efficacy in a murine thrombosis model.

## 2. Materials and Methods

### 2.1. Materials

EPA-EE (≥96%) and NK (20,000 FU/g) were purchased from the Science and Technology Development Center of Zhejiang Provincial Institute of Ocean Development and Xi’an Tianguangyuan Biotechnology Co., Ltd. (Xi’an, China), respectively. Antarctic krill oil (AKO) was provided by Liaoning Fishery Group Co., Ltd. (Donggang, China) Egg yolk lecithin was purchased from Henan Ruilunte Biotechnology Co., Ltd. (Zhenzhou, China) Acetic acid, trichloromethane, sodium thiosulfate titration solution, thiobarbituric acid, trichloroacetic acid, hydrochloric acid, petroleum ether and agarose (1000 U/g) were purchased from Sinopharm Chemical Reagent Co., Ltd. (Shanghai, China), while thrombin and bovine fibrinogen were procured from Beijing Solarbio Science & Technology Co., Ltd. (Beijing, China).

Shanghai Elion Biotechnology Co., Ltd.: Thrombin Time (TT) Assay Kit (MLS03350), Prothrombin Time (PT) Assay Kit (MLS03070), Mouse Fibrinogen (FIB) Assay Kit (ml065212-J), Activated Partial Thromboplastin Time (APTT) Assay Kit (MLS03060), Mouse Tumor Necrosis Factor α (TNF-α) Assay Kit (ml002095-J), Mouse Interleukin 6 (IL-6) ELISA Assay Kit (ml098430-J), Mouse Interleukin 1β (IL-1β) Assay Kit (ml301814-J).

Nanjing has established a Biotechnology Engineering Research Institute: Hydrogen peroxide enzyme (CAT) assay kit (A007-1-1), Malondialdehyde (MDA) assay kit (A003-1-2), High-density lipoprotein cholesterol (HDL-C) assay kit (A112-1-1), Low-density lipoprotein cholesterol (LDL-C) assay kit (A113-1-1), Total superoxide dismutase (T-SOD) assay kit (A015-3-1).

Carrageenan (C1013) and dipyridamole were purchased from Sigma Company (Shanghai, China) and Xiangyu Le Kang Co., Ltd. (Heze, China), respectively. 4% paraformaldehyde (P0099) was obtained from Beyotime Biotechnology Co., Ltd. (Shanghai, China).

### 2.2. Preparation of EPA-NK Nanoemulsions

In this study, nanoemulsions loaded with NK and EPA were prepared by high-speed homogenization (HSH) for coarse emulsification and high-pressure homogenization (HPH) [26] for emulsification. AKO was mixed with Egg yolk phosphatidylcholine (EPC) at a mass ratio to obtain a compound emulsifier. This emulsifier was added to the EPA and stirred at room temperature. The process continued until the emulsifier was fully dissolved in EPA, yielding the EPA mixed phase as the dispersant for later use. Subsequently, the aqueous phase containing NK was added to the oil phase at a volume ratio of 5:1, and coarse emulsion was obtained by high-speed homogenization (8000 r/min, 6 min) using an IKA stirrer (RW20, IKA GmbH, Staufen, Germany). The coarse emulsion was further emulsified and homogenized using a high-speed homogenizer (GA-10H, Shanghai Langhao Fu Nano Technology Co., Ltd., Shanghai, China) to obtain the nanoemulsion loaded with EPA-NK. During the homogenization process, a cold trap was used to cool the homogenization valve to prevent the influence of high temperature on the active substances.

#### 2.2.1. Preparation of EPA-NK Nanoemulsion by Single Factor Test

Single-factor experiments were conducted to investigate the significant effects of emulsifier dosage, emulsifier blending ratio, homogenization pressure and homogenization times on the stability of the prepared EPA-NK nanoemulsion. The standard conditions were fixed as follows: NK (20,000 U/g) enzyme dosage of 1.5 g, EPA dosage of 20%, shear speed of 8000 r/min, and shear time of 6 min. The effects of emulsifier dosage (1%, 1.5%, 2%, 2.5%, 3.0%, m/m), emulsifier blending ratio AKO: EPC (0:6, 1:5, 2:4, 3:3, 4:2, 5:1, 6:0, m/m), homogenization times (2, 3, 4, 5, 6 times), and homogenization pressure (60, 70, 80, 90, 100 MPa) on the activity and characteristics of the prepared EPA-NK nanoemulsion were explored.

#### 2.2.2. Optimization of EPA-NK Nanoemulsion by Response Surface Methodology (RSM)

The preparation process of the EPA-NK nanoemulsion was systematically optimized. Initially, a series of single-factor experiments were conducted to identify critical formulation and process parameters and to determine their approximate optimal ranges. Based on the findings from these preliminary studies, RSM was employed for further systematic optimization. The primary objective of this optimization was to maximize the retention rate of enzyme activity, which was selected as the response value. A Box–Behnken Design (BBD) was specifically chosen for this purpose. Four independent variables were investigated: emulsifier dosage (A), emulsifier blending ratio (B), number of high-pressure homogenization cycles (C), and homogenization pressure (D). Each factor was examined at three coded levels: −1 (low), 0 (medium), and +1 (high), with the specific values for these levels detailed in Appendix A. This experimental design allowed for a comprehensive analysis of the main, interactive, and quadratic effects of the variables on the enzyme activity retention rate (Y).

#### 2.2.3. NK Enzyme Activity Assay

The NK enzyme activity in the nanoemulsion was calculated using the previously reported method [27] with some modifications. At 37 °C, 10 mL of the prepared bovine fibrinogen solution (1.5 mg/mL) was taken in a beaker and stirred while adding 10 mL of agarose solution (15 mg/mL). Then, 1 mL of thrombin solution (10 U/mL) was added to the above mixture, thoroughly mixed, and immediately poured into a 9 mm disposable Petri dish. After being left at room temperature for 1 h, a fibrin plate was prepared and punched for use. After the room temperature was balanced, 10 μL of the sample was spotted in each hole. After the sample holes were spotted, the fibrin plate was placed in a constant temperature and humidity incubator at 37 °C for 18 h. Following the reaction, the diameter of the transparent dissolution circle around the hole was measured with a caliper to assess the fibrinolytic activity of the sample. Urokinase was employed as a standard thrombolytic agent to create a standard curve correlating the diameter of the transparent circle with fibrinolytic activity. According to this curve, the diameter was linearly related to the logarithm of the fibrinolytic activity.

#### 2.2.4. EPA Embedding Rate Determination

Mix 10 mL of EPA-NK nanoemulsion with 15 mL of petroleum ether and shake for one minute. Then, let it naturally stratify. Repeat the extraction three times to ensure thorough extraction and collect the upper organic phase in a flask. Weigh the samples and dry them in a hot air oven (DGG-9140A, Shanghai Senxin Experimental Instrument Co., Ltd., Shanghai, China) at 60 °C until a constant weight is achieved. After cooling, weigh it again and calculate the surface oil content based on the mass change. The encapsulation rate of EPA is calculated according to the following formula: EPA embedding rate (%) = [1 − (Surface oil/total oil)] × 100.

#### 2.2.5. Particle Size, Zeta Potential, and Polydispersity Index (PDI) Determination

The average particle size, PDI and Zeta potential of the emulsion were determined using the Zetasizer Zano ZS instrument (BENano 90Zeta, Dandong Better Instrument Co., Ltd., Dandong, China) according to the method reported by Khalid et al. [28]. To minimize the multiple light scattering effect, the EPA-NK nanoemulsion was diluted 500 times with deionized water prior to measurement. The parameters were set as follows: refractive index = 1.33, equilibration time 120 s.

### 2.3. Physicochemical Properties

#### 2.3.1. Enzyme Activity Assay

An enzyme activity assay was performed under conditions identical to Section 2.2.3, using the same experimental instruments.

#### 2.3.2. Particle Size, PDI and Zeta Assay

The particle size, PDI and Zeta assays were performed under identical conditions to Section 2.2.5, employing the same Zetasizer Zano ZS instrument (BENano 90Zeta, Dandong Better Instrument Co., Ltd.).

#### 2.3.3. Microstructure Analysis

A 10 μL sample of EPA-NK nanoemulsion was placed on a glass slide and covered with a cover glass. It was then observed under an optical microscope (E100, Nikon, Tokyo, Japan) using a 40× objective lens, and the microscopic structure of the emulsion was documented.

#### 2.3.4. Stability Under Different Environmental Stresses

Take 10 mL of the EPA-NK nanoemulsion sample in a test tube and use the untreated sample as a blank control for comparison. Then, adjust the pH, temperature and NaCl concentration of the sample groups, respectively to determine their stability under the corresponding conditions. In addition, the centrifugal stability of the EPA-NK nanoemulsion was evaluated by the centrifugal acceleration method.

pH

The pH value of the emulsion was measured using a pH meter (PHS-3C, Shanghai Hongyi Instrument & Meter Co., Ltd., Shanghai, China). The initial pH value of the EPA-NK nanoemulsion was 7.2 ± 0.2. The pH value of the samples was adjusted to 2.0, 3.0, 4.0, 5.0, 6.0 and 8.0 using 0.1 mol/L HCl or 0.1 mol/L NaOH. Then, the samples were balanced at room temperature for 30 min. After the balance was completed, the pH was adjusted to neutral and characterized.

Temperature

The EPA-NK nanoemulsion was treated at different temperatures: 30 °C, 40 °C, 50 °C, 60 °C, 70 °C, and 80 °C for 30 min, respectively, and then equilibrated to room temperature for characterization and determination.

NaCl

Different concentrations of NaCl solution were added to the EPA-NK nanoemulsions of the same volume, respectively, and the NaCl concentrations were adjusted to 100, 200, 300, 400 and 500 mmol/L. The samples were balanced at room temperature for 2 h and then characterized and measured.

Physical Stability (Ke)

Based on the reported method [29] and with slight modifications, the centrifugal stability of EPA-NK nanoemulsion was determined by the centrifugal acceleration method. The EPA-NK nanoemulsion was diluted 50 times with deionized water and centrifuged at 2000 g for 30 min. After centrifugation, equal amounts of samples were taken from the same position at the bottom layer, and the absorbance values before and after centrifugation were measured at a wavelength of 500 nm using an enzyme-linked immunosorbent assay reader (SynergH, Hangzhou Zhonghe Biotechnology Co., Ltd., Hangzhou, China). The calculation formula for Ke of the EPA-NK nanoemulsion is as follows: Ke = [(A − A_0_)/A] × 100. In the formula, A and A_0_ are the absorption values of the EPA-NK nanoemulsion before and after centrifugation, respectively.

### 2.4. Oxidative Stability

Based on the method of Sharma et al. [30] with slight modifications, the oxidative stability of EPA-NK nanoemulsion was determined. The EPA-NK nanoemulsion was transferred into vials and exposed to rapid oxidation within a stable temperature incubator at 40 °C for 7 days. Throughout this phase of accelerated oxidation, daily samples were collected to assess alterations in POV and TBARs levels, thus evaluating the oxidative resilience of the EPA-NK nanoemulsion.

#### 2.4.1. POV Content Determination

Refer to the first titration method in GB5009.227-2016 [31] for determination, the calculation formula is as follows:$$POV=\frac{\left(V-V_{0}\right)\times C}{2\times m}\times 1000$$

In the formula: POV—the peroxidation value of the grease, mmol/kg;

V—the volume of sodium thiosulfate consumed by the sample, mL;

V_0_—empty sodium thiosulfate consumed by volume, mL;

C—the concentration of sodium thiosulfate standard solution, mol/L

m—sample quality, g;

1000—conversion factor.

#### 2.4.2. TBARs Determination

The TBARs content determination was adapted from a previously reported method [32]. Specifically, 0.2 mL of EPA-NK nanoemulsion was mixed with 1.8 mL of deionized water and 4 mL of a thiobarbituric acid reagent (containing 15 g of trichloroacetic acid, 0.375 g of TBA, 1.76 mL of 12 M hydrochloric acid, and 82.9 mL of deionized water) in a 15 mL centrifuge tube. This mixture was heated at 90 °C for 15 min. After cooling to room temperature, the mixture was centrifuged at 3500 g for 15 min. The absorbance of the supernatant was measured at 532 nm. The TBARs content in the samples was accurately determined using a calibrated 1,1,3,3-tetraethoxypropane standard curve.

### 2.5. Biological Activity

#### 2.5.1. Animals and Diets

Seventy six-week-old male BALB/c mice were selected and acclimated for one week before the formal experiment. The mice were divided into 7 groups (10 mice per group). The grouping and test samples administration details are shown in Table 1. The experimental period was 9 days. All test samples were administered to the mice by gavage. On the 7th day, 2 h after test samples administration, each group of mice was intraperitoneally injected with 50 mg/kg bw of carrageenan solution (prepared freshly). In contrast, the blank group was injected with an equal volume of normal saline. After injecting carrageenan/normal saline, all animals were placed in an environment of 18 ± 1 °C for acute modeling [33]. Test samples administration continued for 2 days after modeling, during which the mice were allowed to eat normally. Dissection was performed on the 9th day. The animal study was carried out in accordance with the Guidelines for the Care and Use of Laboratory Animals of China and was approved by the Animal Care and Use Committee of Zhejiang Ocean University (Animal ethics number: 2023077).

#### 2.5.2. Weight and Black-Tail Length

Record the daily weight changes of mice and observe and measure the length of black blood clots at the tail of mice in each group after intraperitoneal injection of carrageenan for modeling (24 h and 48 h).

#### 2.5.3. Serum and Tissue Sample Analysis

After the animal experiments were completed, all the mice were fasted but not deprived of water overnight after the last administration. The mice had their eyeballs removed and blood was collected for blood sample collection. The blood samples were added to sodium citrate anticoagulant at a volume ratio of 9:1, gently mixed, left at room temperature for 2 h, centrifuged at 2500 g for 15 min (TD5K, Changsha Dongwang Testing Instrument Co., Ltd., Changsha, China), and the upper plasma was taken immediately for the determination of four coagulation indicators.

After collecting the blood samples, the mice were euthanized by dislocation of the cervical vertebrae and their liver, kidney and tail tissues were collected and gently washed with pre-cooled normal saline. The tissues were dried with filter paper and wiped clean with filter paper to remove the buffer solution. The wet weights of the liver and kidney were recorded and then stored at −80 °C for subsequent analysis. After removing the tail tissue, the black tail formation rate within 48 h was recorded, and then the tissue was rinsed with normal saline and fixed in 4% paraformaldehyde overnight.

#### 2.5.4. Organ Index Determination

Blood was collected from mice for dissection. The livers, spleens and kidneys of mice were separated and placed in physiological saline for simple rinsing. The water was then absorbed with filter paper and weighed. The results were recorded. The organ index was calculated according to the formula.Organ index (%) = (Organ weight/Mouse weight) × 100

#### 2.5.5. Prothrombin Time (PT) Determination

Take 70 μL of the plasma obtained above and 140 μL of the PT thromboplastin solution and place them, respectively in the 37 °C pre-warming zone for accurate pre-warming for 180 s. After the incubation, quickly mix the plasma to be tested with the PT thromboplastin solution and add the test beads. Then, the coagulometer (LG-PABER-I, Beijing Shidi Scientific Instrument Co., Ltd., Beijing, China) will be immediately started for testing and recording the results, which is the value of PT.

#### 2.5.6. APTT Determination

Take 50 μL of the plasma obtained above and 50 μL of APTT reagent, mix the mixture thoroughly, add the test beads, and place it in the 37 °C pre-preheating zone for accurate pre-warming for 180 s. At the same time, also preheat the CaCl_2_ solution in the 37 °C preheating zone. After the incubation, quickly add 50 μL of CaCl_2_ solution to the solution to be tested, immediately start the coagulometer for testing and record the result, which is the value of APTT.

#### 2.5.7. TT Determination

Take 50 μL of the obtained plasma and place it in a 37 °C pre-warming zone for an accurate pre-warming duration of 180 s. After incubation, mix the test plasma with thrombin solution quickly, add test beads, initiate the coagulometer for testing immediately, and record the results as the TT (Thrombin Time) value.

#### 2.5.8. FIB

The FIB was determined according to the instructions of the mouse fibrinogen (FIB) kit.

#### 2.5.9. Biochemical Measurement

The levels of oxidative stress in mouse liver homogenates were determined using different kits: catalase (CAT), total superoxide dismutase (T-SOD), and malondialdehyde (MDA) content; inflammatory factors: interleukin-1β (IL-1β), tumor necrosis factor-α (TNF-α), and interleukin-6 (IL-6) levels. Additionally, changes in high-density lipoprotein (HDL-C) and low-density lipoprotein (LDL-C) content were evaluated. The specific procedures were conducted according to the kit instructions.

#### 2.5.10. Histopathological Examination

At the end of the experiment, the appearance of the mouse tails was observed, and the length of the black tail formation was measured and recorded macroscopically. The liver tissues were cut, fixed in 4% paraformaldehyde overnight, and then paraffin-embedded and sectioned at a thickness of 5 μm. Subsequently, hematoxylin and eosin (H&E) staining was performed routinely. Tail vascular tissue sections must be decalcified for one month before paraffin embedding and sectioning. The pathological damage to the liver tissues and tail vessels in each group of mice was observed and compared.

### 2.6. Statistical Analysis

All experimental data are presented as mean ± standard deviation (SD) (*n* = 10). Statistical significance was assessed using SPSS 20.0 software, and differences among means were evaluated using Duncan’s multiple range test. *p* value < 0.05 was deemed statistically significant. Histopathological section analysis was conducted using K-Viewer ((1.5.3.1)) software.

## 3. Results and Discussion

### 3.1. Results of Single-Factor Experiments for the EPA-NK Nanoemulsion

#### 3.1.1. Emulsifier Addition Amount

The accurate control of emulsifier dosage has a significant impact on the strength of the interfacial film formed on the surface of the dispersed phase droplets [34]. It is an indispensable factor in optimizing emulsion stability [35]. As the emulsifier dosage increases, the NK enzyme activity retention rate in the nanoemulsion shows an upward trend and gradually stabilizes (Figure 1A). Both the EPA embedding rate and the absolute value of the Zeta potential show a significant positive increase with the emulsifier dosage (Figure 1B,D). At the same time, the particle size significantly decreases (Figure 1C). When the emulsifier dosage increases from 1.0% to 3.0%, the emulsion particle size and Zeta potential decrease by 87.50 nm and 12.73 mV, respectively, indicating that the increase in emulsifier dosage significantly improves the stability of the emulsion [36]. This is because when the emulsifier dosage increases, the oil-water interfacial tension decreases, resulting in the formation of smaller ultrafine droplets [37]. When the emulsifier dosage is between 2.0% and 3.0%, the emulsion NK activity retention is relatively high, and the EPA embedding rate is also high, with relatively high emulsion stability (smaller particle size and larger absolute value of Zeta potential). Therefore, emulsifier dosages of 1.5%, 2.0%, and 2.5% were selected for subsequent optimization experiments.

Emulsifiers play a crucial role in ensuring the physical and chemical stability of nanoemulsions. They influence the size of the droplets formed during the homogenization process, as well as the stability of the resulting emulsion. Two or more emulsifiers or a combination of emulsifiers and emulsifying aids are often used to suit various special purposes. Based on preliminary experiments, the optimal compound emulsifier selected was the phospholipid emulsifier AKO and EPC. AKO is a dark red oily substance extracted from Antarctic krill, and both AKO and EPC are rich in phosphatidylcholine (PC). A large amount of PC can promote the formation of O/W type emulsions [38] and has a good emulsifying effect.

As the blending ratio of AKO: EPC increases, the enzyme activity of the emulsion first increases and then decreases. At a blending ratio of 4:2, the NK enzyme activity retention rate is the highest (Figure 1E). Compared with a single EPC, adding AKO can significantly reduce the particle size and PDI value of the EPA-NK nanoemulsion (Figure 1F), making the emulsion more stable and the droplet distribution more concentrated. Considering all the indicators comprehensively, the formula with a higher retention rate of active ingredients and stable emulsion is selected. When the AKO: EPC ratio is 3:3, 4:2, and 5:1, the particle size and PDI of the emulsion are relatively low, the emulsion is more stable, and the NK enzyme activity retention rate and EPA embedding rate are higher. Therefore, the above three blending ratios are selected for the response surface optimization test in the subsequent process.

#### 3.1.2. Homogenization Times and Pressure

Figure 2 shows that raising homogenization cycles from 2 to 6 progressively decreases NK activity retention rate (from 74.03% to 38.55%), particle size and PDI. The reason may be that the mechanical energy is converted into thermal energy during the high-pressure homogenization process, and the instantaneous high temperature generated by the homogenizer leads to the gradual loss of NK enzyme activity. As the homogenization times increase, the particle size and PDI of the EPA-NK nanoemulsion decrease significantly. However, when the homogenization times increase from 5 to 6, the particle size of the emulsion shows an upward trend, which is consistent with the research results of Li et al. [39]. The reason may be that excessive homogenization treatment during the homogenization process destroys the nanoemulsion structure system and reduces the stability of the newly formed nanoparticles [40]. In the study of Hamed et al. [41], when the homogenization times are higher than 6, although the particle size of the emulsion does not increase, the downward trend becomes insignificant. When the homogenization times are more, the rupture of emulsion droplets and the leakage of embedded substances lead to the decrease in EPA embedding rate. The highest embedding rate of 98.76% is obtained when homogenized 3 times. It is worth noting that the Zeta potential of the EPA-NK nanoemulsion does not show significant changes under different homogenization times. Therefore, 2–4 times of high-pressure homogenization are selected.

With the increase in homogenization pressure, the enzyme activity retention rate of EPA-NK nanoemulsion showed a significant downward trend (Figure 2E), decreasing from 90.10% to 43.68%. The reason for this might be the influence of instantaneous high temperature during high-pressure homogenization, and the high pressure could also cause protein denaturation, which might be a potential factor for the loss of NK enzyme activity. The increase in homogenization pressure led to a notable decrease in the particle size of EPA-NK nanoemulsion (Figure 2G), which aligns with prior research [42]. The possible reason is that the shear force from high-pressure homogenization [28] broke down the structure of larger droplets. This action reduced the particle size and enhanced the stability of the emulsion. Additionally, the embedding rate of EPA was higher. However, it should be noted that when the homogenization pressure exceeded 90 MPa, the embedding rate of EPA decreased (Figure 2F). The rise in pressure also generated more heat energy, which might lead to demulsification in nanoemulsion systems inherently in an unstable thermodynamic equilibrium. This, in turn, affects the stability of the emulsion [43]. Therefore, the high-pressure homogenization pressure was selected to be 70–90 MPa.

### 3.2. The Results of the Response Surface Optimization Experiment

#### 3.2.1. Establishment of Regression Equation Model and Significance Test

Response-surface analysis (Appendix A) yielded a highly significant model (*p* < 0.0001) with non-significant lack of fit (*p* = 0.1273), indicating that only random error contributed to residuals. R^2^ = 0.9711 and R^2^adj = 0.9422 show that 97.11% of the variance in NK activity retention was explained by the selected factors, while CV = 3.32% confirms high precision. Thus, the fitted regression equation reliably predicts EPA-NK nanoemulsion enzyme activity retention within the design space.

Enzyme activity retention rate (%) = 90.48 + 0.625A − 1.26B − 3.16C − 2.06D + 0.755AB − 4.32AC − 2.56AD + 4.12BC − 1.9BD − 1.5CD − 6.93A^2^ − 6.7B^2^ − 10.85C^2^ − 17.31D^2^

In the above equation: A. amount of emulsifier (%); B. emulsifier compounding ratio (AKO: EPC); C. high pressure homogenization times (times); D. high pressure homogenization pressure (MPa).

Four factors influence the enzyme activity retention rate of EPA-NK nanoemulsion in the order of influence: homogenization times > homogenization pressure > emulsifier ratio (AKO: EPC) > emulsifier addition amount. Nanoemulsion cannot form spontaneously and requires the input of external energy and the stabilization of emulsifiers. When the ratio and content of emulsifiers reach an appropriate level, the two necessary processing conditions of homogenization times and homogenization pressure become the primary determinants of the system’s formation.

In the variance analysis of the enzyme activity retention rate model, the *p*-values of C, AC, BC, A^2^, B^2^, C^2^, and D^2^ are all less than 0.01. This indicates that these factors have an extremely significant impact on the enzyme activity retention rate. The results show that there is no simple linear relationship between homogenization time and emulsifier ratio and addition amount.

The *p*-value of homogenization pressure D is all less than 0.05, indicating a significant influence on the enzyme activity retention rate, while the differences in other items are not significant.

#### 3.2.2. Response Surface Factor Interaction Analysis

The three-dimensional response-surface plots (Figure 3) reveal how enzyme activity retention varies with the tested factors. The steeper the 3D graph, the more significant the interaction effect of the factors on the enzyme activity retention rate. The dense contour lines and steep surfaces for the AC and BC interactions confirm that homogenization cycles and emulsifier dosage, as well as the AKO:EPC ratio, exert the most pronounced combined influence on retention—consistent with the ANOVA results.

#### 3.2.3. Determination and Verification of Optimal Process Conditions

Through the optimization analysis of the model regression equation, the optimal process conditions for preparing EPA-NK nanoemulsion are determined as follows: emulsifier addition amount of 2.053%, blending ratio (AKO:EPC) of 5:3, homogenization times of 2.81 times, and homogenization pressure of 79.486 MPa. Under these conditions, the theoretical predicted value of the enzyme activity retention rate is 90.95%. For the convenience of practical operation, the predicted conditions are modified as follows: emulsifier addition amount of 2.0%, blending ratio (AKO:EPC) of 5:3, homogenization times of 3 times, and homogenization pressure of 80 MPa. Under these conditions, the verification experiment is conducted, and the particle size of the obtained EPA-NK nanoemulsion is 300.25 ± 6.84 nm, the EPA embedding rate is 98.41 ± 0.52%, and the measured value of the enzyme activity retention rate is 90.26 ± 0.64%. The predicted and measured values are within the confidence interval, indicating that the experimental value of the enzyme activity retention rate is in good agreement with the predicted value of the regression equation. Therefore, the regression equation of the enzyme activity retention rate of EPA-NK nanoemulsion obtained through the Design-expert.V8.0.6.1 program analysis is effective.

### 3.3. Test Results on the Effects of Different Environmental Stresses on the Stability of EPA-NK Nanoemulsion

The changes in the activity and characteristics of EPA-NK nanoemulsions after treatment at different pH levels are shown in Figure 4A. When the pH is within the range of 5 to 8, the stability of EPA-NK nanoemulsion is relatively high. When the pH is lower than 5, the stability of EPA-NK nanoemulsion significantly decreases as the pH drops, and the NK activity retention rate also significantly declines with the decrease in pH value.

Emulsions may be affected by different temperature conditions during manufacturing, transportation and storage. For instance, high temperatures can cause the components of emulsions to stratify, solidify or deteriorate, while low temperatures may lead to freezing or crystallization [44]. By exploring the temperature stability of emulsions, the stability of emulsions under various temperature conditions can be determined, thereby ensuring the quality of emulsions. Under the temperature range of 30 to 60 °C, EPA-NK nanoemulsion can maintain its stability well. When the temperature exceeds 60 °C, its stability significantly decreases, and the NK activity is lost relatively quickly.

Salt ions can influence emulsion colour, texture, and physicochemical stability; therefore, their effect on EPA-NK nanoemulsions was systematically examined. As shown in Figure 4E, the NK enzyme activity retention rate of EPA-NK nanoemulsion did not show significant changes among different NaCl concentrations, indicating that different salt ion environments do not affect the NK activity of EPA-NK nanoemulsion. When the salt ion concentration gradually increased to 200 mmol/L, the particle size of EPA-NK nanoemulsion did not show significant changes. However, as the salt ion concentration continued to increase, the particle size of EPA-NK nanoemulsion showed a significant increasing trend. This transition can be attributed to the infiltration of salt ions into the electrical double layer, which neutralizes surface charges, reduces charge density, and thereby increases the absolute value of the Zeta potential. This phenomenon is called the charge shielding effect [45]. The microscopic structure of EPA-NK nanoemulsion changed with the salt ion concentration, as shown in Figure 4F. It can be seen from the figure that there was no obvious change in the microscopic structure of EPA-NK nanoemulsion when the salt ion concentration was between 100 and 300 mmol/L.

### 3.4. EPA-NK Nano Emulsion Oxidation Stability Test Results

Oxidative stability is one of the key factors affecting the quality of emulsions. Studying the oxidative stability of emulsions can help identify potential oxidation problems early and find innovative solutions for improvement to enhance the quality and stability of emulsions and meet consumer demands [46]. In this study, EPA-NK nanoemulsions were stored at 40 °C for 7 days, and the oxidative stability of EPA-NK nanoemulsions was evaluated by measuring the contents of primary oxidation products (POV) and secondary oxidation products (TBARs). As shown in Figure 5, the initial POV values of both EPA-NK nanoemulsions and unencapsulated EPA were relatively low. Over time, the POV and TBARs values of both EPA-NK nanoemulsions and unencapsulated EPA showed a significant upward trend. However, the growth trends of POV and TBARs contents of unencapsulated EPA were significantly higher than those of EPA-NK nanoemulsions. Moreover, the POV and TBARs contents of the emulsions were significantly lower than those of unencapsulated EPA at different storage days. The results indicate that the nanoemulsion encapsulation system can protect EPA and effectively inhibit or slow down the oxidation of EPA.

### 3.5. Interventional Effect of EPA-NK Nanoemulsion on Thrombosis in Mice

#### 3.5.1. Safety Assessment

As shown in Appendix A, before the treatment with the carrageenan, the body weights of all groups of mice did not show significant changes. This indicates that the administration of the EPA-NK nanoemulsion sample did not affect the body weight of the mice. After the treatment with the carrageenan, the body weights of all groups of mice decreased, and the daily diet decreased, while the water intake increased. The reason for this might be the pathological state of the mice and the low-temperature environment they were in.

During the 9-day gavage administration period, no toxicological deaths occurred in any experimental group. Following cervical dislocation, organ indices are presented in Table 2. The experimental results demonstrate: (1) no significant differences in hepatic indices were observed among all groups; (2) splenic indices showed no significant intergroup variations except for the blank control group; (3) renal indices exhibited moderate improvement following EPA-NK nanoemulsion administration.

#### 3.5.2. Thrombosis Inhibition Effect

After mice were injected with carrageenan, thrombi formed in their tails and developed over time. The EPA-NK nanoemulsion intervention significantly inhibited the progression of thrombosis (the dark area indicated by the red line segment in Figure 6A). The quantitative results of the thrombosis formation rate (the ratio of the length of the black tail to the total tail length) showed that the 48 h black tail formation rate of the model group mice (93.6 ± 0.9%) was significantly higher than that of the positive drug dipyridamole (32.5 ± 2.3%) and the EPA-NK nanoemulsion sample groups. The black tail formation rate of the EPA alone group (94.7 ± 1.6%) was not significantly different from that of the model group. The black tail formation rate of the high-dose EPA-NK nanoemulsion (64.2 ± 3.8%) was significantly lower than that of the low and medium doses. Therefore, it can be concluded that the EPA-NK nanoemulsion samples have a certain inhibitory effect on the black tail thrombosis caused by carrageenan in mice, and the high-dose EPA-NK nanoemulsion has the best effect.

Histopathology directly reflects morphological changes in tissues. Through H&E staining of mouse tails and livers, the preventive effect of EPA-NK nanoemulsion on carrageenan-induced thrombosis formation can be observed [47]. As shown in Figure 6B,C, the tail vein vessels of the blank control group mice were intact and smooth, with a well-defined structure. In contrast, the model group exhibited vascular wall inflammation, leukocyte infiltration, bleeding, and obvious thrombosis within the vessels. EPA-NK nanoemulsion and dipyridamole were both effective in alleviating the venous thrombosis in mice.

The results at 5 cm from the tail tip were better than those at 3 cm for all groups. This is because the black-tail thrombosis in mice forms gradually from the tail tip towards the base, leading to more severe vascular lesions and thrombosis closer to the tail tip. The thrombosis may still be in the formation stage at a greater distance from the tail tip, resulting in less severe vascular damage. No obvious liver damage was observed in the liver tissue sections of all groups (Figure 6D). To sum up, the findings indicate that the EPA-NK nanoemulsion can prevent thrombosis in mice.

#### 3.5.3. EPA-NK Nanoemulsion Modulates Coagulation Parameters and Plasma Lipoprotein Profiles in Mice

The four items of coagulation (PT, APTT, TT, FIB) are the routine methods for detecting abnormalities in the coagulation system in clinical practice, and they can accurately detect blood diseases [48].

As shown in Figure 7A, the PT time of the blank group was significantly higher than that of the other groups (10.0 ± 1.9 s), and the PT time of the model group (5.6 ± 0.3 s) was significantly lower than that of each sample treatment group. The significant shortening of PT time in mice is attributed to their pathological state, where thrombi are continuously generated. This condition may cause abnormal activation or increased synthesis of coagulation factors. Consequently, this leads to a marked reduction in PT time.

During the process of thrombosis, a significant deficiency of coagulation factors can also cause the APTT to shorten. The APTT time of the model group (12.8 ± 5.3 s) was significantly lower than that of dipyridamole group (26.1 ± 3.7 s) and the EPA-NK nanoemulsion sample group. The APTT time of the EPA treatment alone group (20.2 ± 3.7 s) was significantly higher than that of the model group, indicating a specific alleviating effect. No significant differences existed among the EPA-NK nanoemulsion sample, dipyridamole, and blank groups (Figure 7B).

The FIB content in the model group (76.5 ± 9.3 g/L) was significantly higher than in the other groups. The EPA-alone treatment group (63.6 ± 6.1 s) showed a significant decrease, but its effect was not as pronounced as that of the sample treatment group. The FIB content in the high-dose group of EPA-NK nanoemulsion samples (31.5 ± 5.3 s) was significantly lower than that in the low-dose group (46.8 ± 5.7 s), and there was no significant difference in FIB content between the medium-dose and high-dose groups of EPA-NK nanoemulsion and dipyridamole group (35.5 ± 5.7 s). The effects of the medium-dose and high-dose groups of EPA-NK nanoemulsion were comparable to those of dipyridamole (Figure 7C).

The TT time in the model group (27.0 ± 6.0 s) was significantly higher compared to dipyridamole group (17.4 ± 1.4 s) and the EPA-NK nanoemulsion sample group. Moreover, there was no significant difference in TT time among the EPA-NK nanoemulsion low-dose group, high-dose group, the EPA-alone group (18.6 ± 2.0 s), and dipyridamole group (Figure 7D).

High-density lipoprotein (HDL) removes excess cholesterol from murine tissues to the liver, lowering atherosclerotic risk [49,50], whereas low-density lipoprotein (LDL) delivers hepatic cholesterol to peripheral tissues; elevated LDL promotes arterial deposition and cardiovascular disease [51]. Monitoring HDL/LDL dynamics therefore gauges murine lipid metabolism and cardiovascular risk [52].

The high-density lipoprotein level in the model group was significantly lower than in the other treatment groups. Meanwhile, the low-density lipoprotein level in the model group was significantly higher than in the different treatment groups. This suggests that mice in the model group were more susceptible to thrombosis. The regulatory effect of EPA-NK nanoemulsion on high and low-density lipoproteins in mice may also be one of the important factors in alleviating the formation of black-tailed thrombosis (Figure 7E,F).

#### 3.5.4. Mechanism Discussion

Oxidative stress accelerates thrombosis via ROS-mediated platelet activation. The antioxidant enzymes SOD and CAT are therefore crucial for thromboprotection [53,54]. As shown in Figure 8A, the SOD content in the model group was significantly lower than that in the blank group. When compared to the model group, the SOD content in dipyridamole group and the medium and high-dose EPA-NK nanoemulsion groups saw a significant increase. However, there was no significant difference in SOD content between the EPA alone group and the model group. These findings suggest that the EPA-NK nanoemulsion can markedly boost the SOD content in mice and strengthen the body’s antioxidant capacity.

CAT is an enzyme scavenger that can facilitate the decomposition of hydrogen peroxide into molecular oxygen and water, providing antioxidant defense for the body [54]. The CAT content in the model group was significantly lower than in the blank group (Figure 8B). Compared with the model group, the CAT content in dipyridamole group and the EPA-NK nanoemulsion medium and high-dose groups was significantly increased. At the same time, there was no significant difference in the CAT content between the EPA alone group and the model group. The results suggest that EPA-NK nanoemulsion can significantly increase the CAT content in mice.

In the body, free radicals cause lipid peroxidation, and the resulting MDA can indirectly reflect the degree of oxidative stress damage to the body. Therefore, MDA levels may be an important indicator of thrombosis [55]. The results show that after the model was established, the MDA content in the model group was significantly higher than that in other groups. The MDA content in the EPA alone group was considerably lower than that in the model group, indicating a specific alleviating effect. The results suggest that EPA-NK nanoemulsion can significantly reduce the MDA content in mice. These results indicate that thrombosis is closely related to oxidative stress. Under different doses of administration, these oxidative-related indicators can be alleviated to a certain extent, and some indicators can be comparable to those of dipyridamole (Figure 8C).

Studies indicate a mutual causal relationship between inflammation and thrombosis, with their interaction exacerbating organ damage and posing life-threatening risks. Specifically, inflammatory stimuli activate Toll-like receptors, which in turn activate immune effector cells (monocytes, macrophages, lymphocytes, endothelial cells) and trigger release of inflammatory cytokines such as TNF-α and IL-6 [56].

TNF-α regulates NF-κB and MAPK cascades, driving IL-6 and IL-1β release that amplifies leukocyte–endothelial injury and thrombus growth [57]. Tight control of TNF-α, IL-1β and IL-6 levels thereby modulates both inflammation and thrombosis-related gene expression, offering a therapeutic target for antithrombotic strategies [58].

The changes in inflammatory factor levels in each group of mice are shown in Figure 8D–F. Compared with the normal group, the levels of TNF-α, IL-6, and IL-1β in the liver tissues of the model group mice were significantly increased. Compared to the model group, the levels of IL-6, TNF-α and IL-1β were reduced considerably in the dipyridamole, EPA, and EPA-NK nanoemulsion sample groups. In the NK-H group, the levels of TNF-α, IL-6, and IL-1β showed no significant difference compared to dipyridamole group (*p* > 0.05). The research results indicate that dipyridamole and EPA-NK nanoemulsion intervened in the above inflammatory cytokines in this study, thereby inhibiting thrombosis.

## 4. Conclusions

This study optimized an EPA-NK nanoemulsion stabilized by a composite emulsifier (AKO and EPC) and systematically evaluated its physicochemical properties, safety, and preliminary in vivo antithrombotic efficacy. The optimal formulation was prepared using 2% total emulsifier (AKO:EPC = 5:3) and three homogenization cycles at 80 MPa, yielding particles with a mean diameter of 300 nm (polydispersity index, PDI = 0.16), ζ-potential of −48.6 mV, NK retention rate of 90%, and EPA encapsulation efficiency of 98%. Physicochemical stability tests confirmed the nanoemulsion maintained structural integrity under physiological conditions (pH 5–8, 30–60 °C), while extreme pH (<5) or heat (>60 °C) reduced stability and NK activity, with salt ions exerting negligible effects.

Toxicological assessment supported the formulation’s safety profile: despite historical concerns about NK-related hematological effects [59], the European Food Safety Authority (EFSA) Panel on Nutrition, Novel Foods and Food Allergens (NDA) approved NK as a novel food under Regulation (EU) 2017/115, confirming safety under intended use conditions [60]. Clinical evidence [61] further demonstrated that NK reduces plasma fibrinogen and coagulation factors VII/VIII without adverse effects on blood lipids, uric acid, or clinical parameters. Additionally, the low AKO dosage (1.25% of the total system) avoided sensory interference, resulting in a white, odorless emulsion suitable for oral administration.

In carrageenan-induced tail-thrombosis mice, EPA-NK nanoemulsion showed no toxicity and dose-dependently decreased tail discoloration and thrombus burden, outperforming EPA alone but remaining slightly less effective than dipyridamole. The nanoemulsion normalized APTT, TT, PT and lowered fibrinogen, indicating restored coagulation balance. It also improved lipid profiles, antioxidant status and inflammatory cytokines, collectively limiting endothelial dysfunction and platelet aggregation. Thus, EPA-NK nanoemulsion safely and effectively inhibits thrombosis by modulating coagulation, oxidative stress and inflammation.

These findings provide preliminary evidence that the AKO/EPC-stabilized EPA-NK nanoemulsion is a physicochemically stable, safe, and effective oral delivery system for antithrombotic therapy, with mechanisms involving coagulation modulation, oxidative stress reduction, and inflammation suppression. However, definitive conclusions regarding oral bioavailability require follow-up studies, including pharmacokinetic analysis to quantify absorption efficiency and systemic exposure, physiological digestion models (e.g., INFOGEST) to validate gastrointestinal stability, and long-term toxicity evaluations in larger animal models. Future work will focus on these aspects to advance translational potential.

## Figures and Tables

**Figure 1 foods-14-03482-f001:**
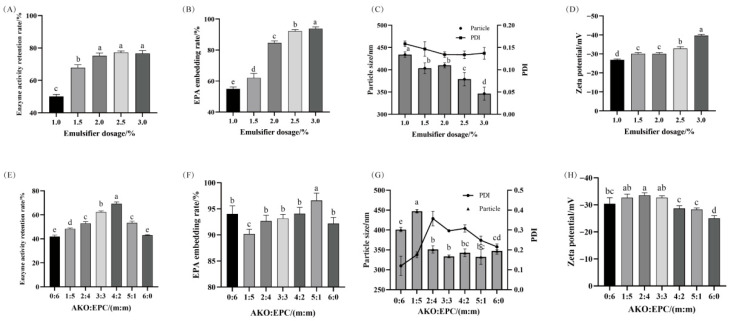
Effect of emulsifier addition mix ratio on EPA-NK nanoemulsions.(**A**) Effect of emulsifier addition level on NK activity retention rate; (**B**) Effect of emulsifier addition level on EPA embedding rate; (**C**) Effect of emulsifier addition level on particle size and PDI; (**D**) Effect of emulsifier addition level on Zeta potential; (**E**) Effect of emulsifier mixing ratio on NK activity retention rate; (**F**) Effect of emulsifier mixing ratio on EPA embedding rate; (**G**) Effect of emulsifier mixing ratio on particle size and PDI; (**H**) Effect of emulsifier mixing ratio on Zeta potential. Different letters (a–e) in (**A**–**H**) represent significant differences among samples (*p* < 0.05).

**Figure 2 foods-14-03482-f002:**
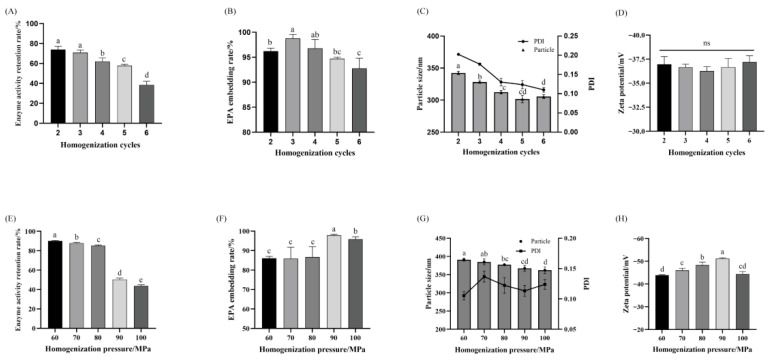
Effect of homogenization cycle times and homogenization pressure on EPA-NK nanoemulsions. (**A**) Effect of high-pressure homogenization cycles on NK activity retention rate; (**B**) Effect of high-pressure homogenization cycles on EPA embedding rate; (**C**) Effect of high-pressure homogenization cycles on particle size and PDI; (**D**) Effect of high-pressure homogenization cycles on Zeta potential; (**E**) Effect of homogenization pressure on NK activity retention rate; (**F**) Effect of homogenization pressure on EPA embedding rate; (**G**) Effect of homogenization pressure on particle size and PDI; (**H**) Effect of homogenization pressure on Zeta potential. Different letters (a–e) in (**A**–**H**) represent significant differences among samples (*p* < 0.05).

**Figure 3 foods-14-03482-f003:**
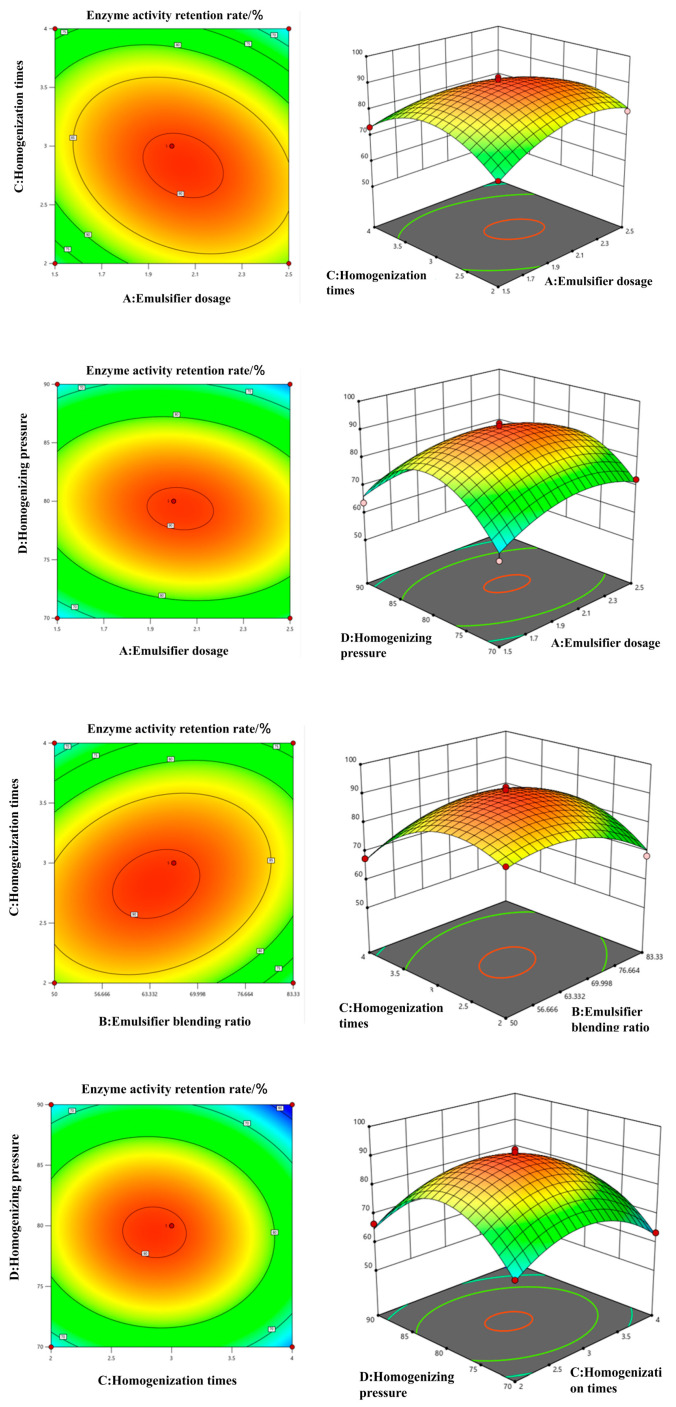
Contour diagram of interaction of each factor and 3D analysis diagram of response surface.

**Figure 4 foods-14-03482-f004:**
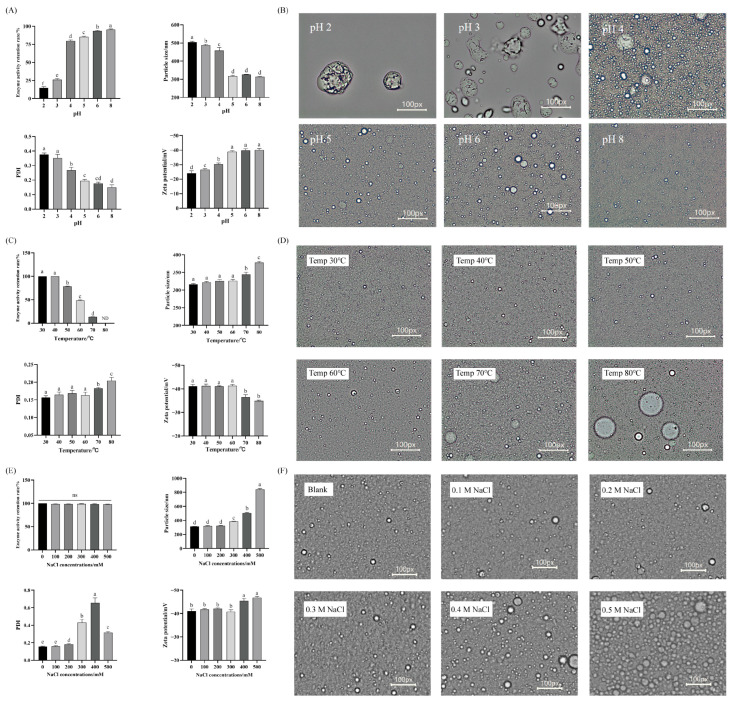
Effect of pH, temperature and salt ion on EPA-NK nanoemulsion. (**A**) Effect of pH on enzyme activity retention rate, particle size, PDI and Zeta potential. (**B**) Different pH values on the macrostructure (pH values from left to right are 2–8). (**C**) Effect of temperature on enzyme activity retention rate, particle size, PDI and Zeta potential. (**D**) Different temperatures on the macrostructure (The salt temperature from left to right is 30–80 °C, respectively). (**E**) Effect of salt ion on enzyme activity retention rate, particle size, PDI and Zeta potential. (**F**) Effect of salt ion on the macrostructure (The salt ion concentrations from left to right are 0–500 mmol/L, respectively). Different letters (a–e) in (**A**,**C**,**E**,**F**) represent significant differences among samples (*p* < 0.05). ‘ns’ denotes “Not Significant,” indicating no statistically significant difference (typically with a *p*-value > 0.05), while ‘ND’ generally stands for “Not Detected.”

**Figure 5 foods-14-03482-f005:**
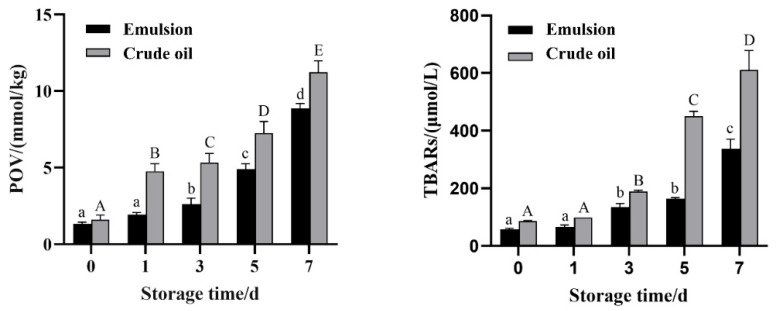
The contents of POV (**left**) and TBARs (**right**) of EPA-NK nanoemulsion changed with accelerated oxidation. Uppercase letters ABCD denote statistical comparisons within the crude oil group, while lowercase letters abcd represent comparisons within the emulsion group; a significance level of *p* < 0.05 is used to indicate statistically significant differences between groups.

**Figure 6 foods-14-03482-f006:**
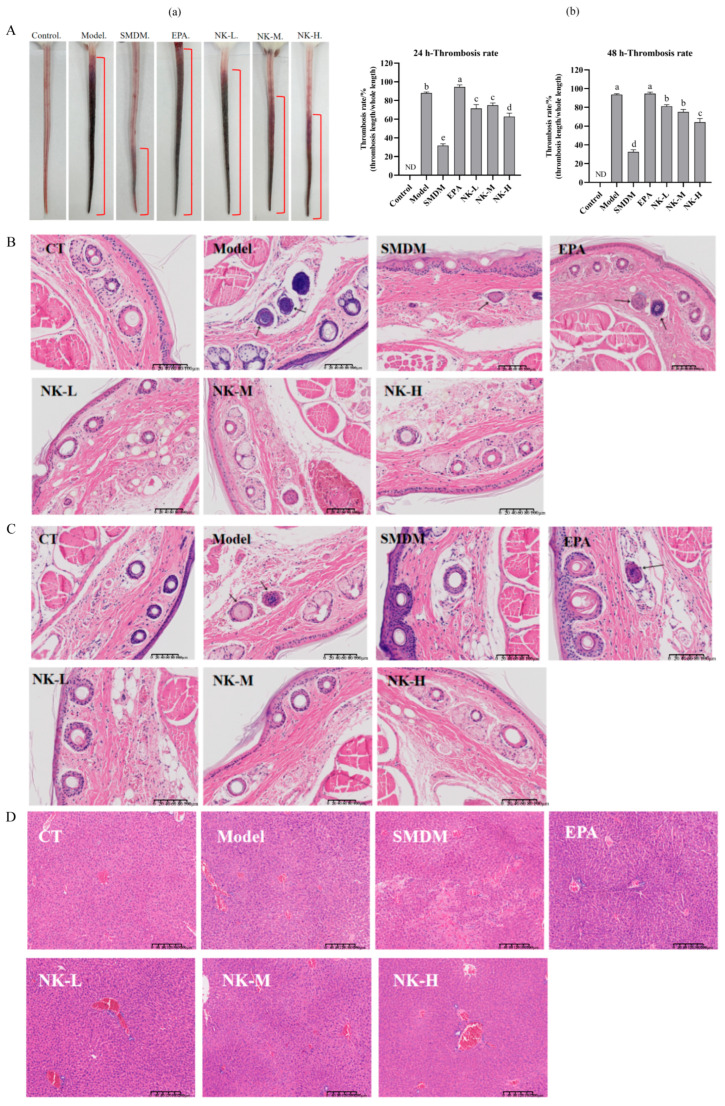
(**A**) Appearance of black-tailed mice 48 h after modeling (**a**), black-tail formation rate of mice in different periods (**b**). (**B**) H&E staining results of mouse tail blood vessels (3 cm from tail tip). (**C**) H&E staining results of mouse tail blood vessels (5 cm from the tail tip). (**D**) H&E staining results of mouse tail blood vessels (5 cm from the tail tip). Different letters (a–e) represent significant differences among samples (*p* < 0.05). And ‘ND’ denotes “Not Detected.”

**Figure 7 foods-14-03482-f007:**
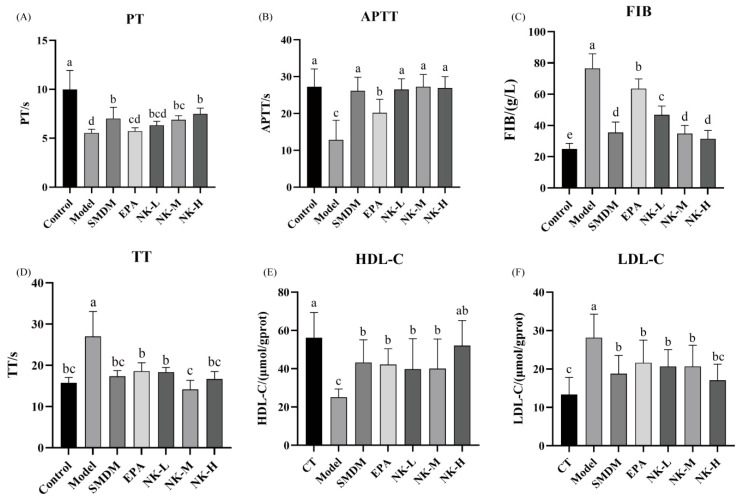
Changes in four indices of blood coagulation and high- and low-density lipoprotein content in mice. (**A**) PT. (**B**) APTT. (**C**) FIB. (**D**) TT. (**E**) HDL-C. (**F**) LDL-C. Different letters (a–e) in (**A**–**F**) represent significant differences among samples (*p* < 0.05).

**Figure 8 foods-14-03482-f008:**
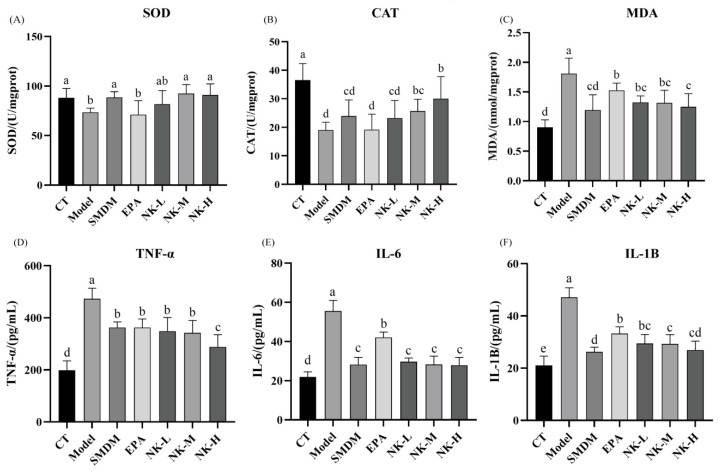
Changes in oxidative stress and inflammatory factors levels in mice. (**A**) SOD. (**B**) CAT. (**C**) MDA. (**D**)TNF-α. (**E**) IL-6. (**F**) IL-1β. Different letters (a–e) in (**A**–**F**) represent significant differences among samples (*p* < 0.05).

**Table 1 foods-14-03482-t001:** Experimental design of the intervention effect of EPA-NK nanoemulsion on black-tail thrombosis in mice.

ID	Group	Administration Details	Modeling on Day 7
A	Blank Control Group	Normal saline	Normal saline
B	Model Group	Normal saline	50 mg/kg bw carrageine
C	Positive Control Group	Dipyridamole	50 mg/kg bw carrageine
D	EPA Control Group	EPA content in the sample medium dose group	50 mg/kg bw carrageine
E	The low-dose EPA-NK nanoemulsion group	4600 U/d (NK enzyme activity)	50 mg/kg bw carrageine
F	The medium-dose EPA-NK nanoemulsion group	6900 U/d (NK enzyme activity)	50 mg/kg bw carrageine
G	The high-dose EPA-NK nanoemulsion group	9200 U/d (NK enzyme activity)	50 mg/kg bw carrageine

**Table 2 foods-14-03482-t002:** Organ index of mice in each group.

Group		Organ Index/%	
	Liver	Kidney	Spleen
Blank control	4.5935 ± 0.23	1.6704 ± 0.08 ^ab^	0.4045 ± 0.05 ^a^
Model control	4.7835 ± 0.19	1.7538 ± 0.06 ^a^	0.5747 ± 0.08 ^b^
Positive control	4.7953 ± 0.43	1.6581 ± 0.10 ^ab^	0.5592 ± 0.05 ^b^
EPA control	4.7268 ± 0.36	1.6833 ± 0.11 ^ab^	0.5770 ± 0.09 ^b^
EPA-NK Low-dose nano-emulsion	4.6533 ± 0.41	1.6336 ± 0.07 ^b^	0.5062 ± 0.08 ^b^
EPA-NK Medium-dose nano-emulsion	4.7316 ± 0.22	1.6655 ± 0.10 ^b^	0.5236 ± 0.05 ^b^
EPA-NK High-dose nano-emulsion	4.8660 ± 0.19	1.7122 ± 0.11 ^ab^	0.5365 ± 0.06 ^b^

Different letters (a,b) indicate significant differences among samples (*p* < 0.05).

## Data Availability

The original contributions presented in this study are included in the article/Appendix A. Further inquiries can be directed to the corresponding authors.

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
