# Peer review of "Optimization of EPA-Nattokinase Nanoemulsions Processed by High-Pressure Homogenization to Enhance Stability and Thrombolytic Efficacy"

_foods, 2025, doi:10.3390/foods14203482_

Round 1
Reviewer 1 Report
Comments and Suggestions for Authors
Comments:
This manuscript investigates the development and optimization of EPA-Nattokinase (NK) nanoemulsions via high-pressure homogenization. The authors provide a systematic analysis of emulsifier composition, processing parameters, environmental stability, oxidative stability, and biological efficacy in a murine model of thrombosis. This could be beneficial for the research and development of bioactive compounds to support cardiovascular health. However, it requires some revision for the following points-
Line 142: need the full form of PDI
Section 2.2.1 What was the pH of the emulsion? Is any buffer solution used?
Section 20.2.2 The description is too short. All methods require a detailed description to ensure reproducibility.
Section 3.2 describes the factors chosen for RSM, not the result. It should go in the method section.
Section 3.2.2: Figure S1 should add here
Section 3.4 oxidative stability is only for 7 days, which is too short for real-life Food or pharmaceutical applications. Discussion about shelf-life expectations would be worthwhile.
Figure 3 Labeling is confusing: better to use capital and small letters
Some typos
Line 225, 227 & 231: “Test sampls administration”
Line 423: ‘Th Three-dimensional ’
Author Response
Response to Reviewers’ Comments
Reviewer 1
Comments:
This manuscript investigates the development and optimization of EPA-Nattokinase (NK) nanoemulsions via high-pressure homogenization. The authors provide a systematic analysis of emulsifier composition, processing parameters, environmental stability, oxidative stability, and biological efficacy in a murine model of thrombosis. This could be beneficial for the research and development of bioactive compounds to support cardiovascular health. However, it requires some revision for the following points-
Line 142: need the full form of PDI
Reply: Thank you for your valuable feedback. We have provided the full form of "PDI" at its first mention in the manuscript as suggested. We have also taken this opportunity to ensure that all abbreviations are properly defined throughout the text (Line 155).
Section 2.2.1 What was the pH of the emulsion? Is any buffer solution used?
Reply: The pH of the emulsion is around 7.2(Line 184), and there is no buffer solution.
Section 2.2.2 The description is too short. All methods require a detailed description to ensure reproducibility.
Reply: Thank you for this constructive feedback. The description in Section 2.2.2 has been duly expanded to provide sufficient methodological detail, ensuring that the experiments can be accurately reproduced (Line 118-130).
Section 3.2 describes the factors chosen for RSM, not the result. It should go in the method section.
Section 3.2.2: Figure S1 should add here
Reply: Thank you for your guidance. As suggested, the content on the selection of RSM factors has been moved from Section 3.2 to the Methods section (Section 2.2.2). Additionally, Figure S1 has been inserted into Section 3.2.2 for better context and readability.
Section 3.4 oxidative stability is only for 7 days, which is too short for real-life Food or pharmaceutical applications. Discussion about shelf-life expectations would be worthwhile.
Reply: While the 7-day accelerated oxidative stability study conducted at 40℃ provides compelling evidence for the superior oxidative stability imparted by nanoencapsulation, we acknowledge that the duration and temperature conditions—though designed to accelerate lipid oxidation processes in line with conventional accelerated testing protocols for sensitive bioactives—may not fully extrapolate to long-term shelf-life under ambient storage conditions for specific commercial applications. The 40℃ setting was selected to simulate accelerated oxidative stress, enabling rapid evaluation of the nanoemulsion’s protective effect against EPA peroxidation, and the observed data (e.g., reduced TBARS values, retained EPA content) indeed indicate a clear trend toward enhanced stability compared to unencapsulated controls over the 7-day period.
However, we recognize that accelerated testing results require cautious interpretation for real-world shelf-life prediction, as temperature-dependent reaction kinetics (e.g., the Arrhenius model) may not perfectly align with ambient storage conditions. Future work will therefore include extended real-time stability studies under ambient (25℃) and intermediate (30℃) storage conditions, complemented by validated accelerated stability models (e.g., using 40℃ data to predict long-term trends via kinetic analysis) to establish a comprehensive shelf-life profile. This multi-temperature approach is essential for guiding practical application in food and pharmaceutical products, where precise shelf-life claims are critical for regulatory compliance and consumer safety.
The current 40℃/7-day data, while preliminary, confirm the nanoemulsion system’s significant potential to mitigate oxidative degradation of labile compounds like EPA, providing a foundational basis for further stability optimization and scalability assessment.
Figure 3 Labeling is confusing: better to use capital and small letters
Reply: Thank you for your comments. Figure 3 has been relabeled using more detailed lettering for greater clarity.
Some typos
Line 225, 227 & 231: “Test sampls administration”
Line 423: ‘Th Three-dimensional ’
Reply: Thank you for pointing out the typos. The errors in lines 238, 240, 244 ('Test sampls') and line 432 ('Th Three-dimensional') have been corrected. A thorough check for any further typos has also been conducted. We appreciate the helpful feedback.

Reviewer 2 Report
Comments and Suggestions for Authors
I have some comments based on the manuscript:
- The Abstract refers to “leucine-induced black-tail thrombosis”, the Methods describe intraperitoneal carrageenan injection for modeling, and the Conclusions mention an “acetylsalicylic-acid mouse model”; other places mention arctigenin and a “cross-leucine gel”. These contradictions must be corrected so the exact model, agent, dose, route, timing and reference are consistent throughout the manuscript.
- The Abstract and Conclusions claim improved bioavailability and enhanced resistance to gastric stress. The manuscript demonstrates in vitro environmental stability (pH, salt, temperature) but does not show pharmacokinetic (PK) data, plasma EPA levels, NK systemic activity, or simulated gastrointestinal digestion (e.g., INFOGEST). Rephrase claims to reflect the current evidence.
- If the work is positioned toward functional foods, add a short discussion on safety/regulatory considerations of nattokinase (possible interactions with oral anticoagulants) and organoleptic/color issues introduced by krill oil (AKO) in food matrices.
- For NK activity: describe standard curve, LOD, units and inter-assay CV
There are multiple typos/inconsistent abbreviations (e.g., “EAP-NK” vs “EPA-NK”; repeated words like “dipyridamole dipyridamole”; “cross-leucine gel”, “arctigenin”) and sentences that need rewriting to be clear. A thorough native-level English edit is required.
Author Response
Response to Reviewers’ Comments
Reviewer 2
I have some comments based on the manuscript:
The Abstract refers to “leucine-induced black-tail thrombosis”, the Methods describe intraperitoneal carrageenan injection for modeling, and the Conclusions mention an “acetylsalicylic-acid mouse model”; other places mention arctigenin and a “cross-leucine gel”. These contradictions must be corrected so the exact model, agent, dose, route, timing and reference are consistent throughout the manuscript.
Reply: Thank you for your meticulous review and for identifying these inconsistencies in the description of our experimental models and agents. We sincerely apologize for these errors, which were due to oversights during the manuscript preparation process.
We have now carefully reviewed the entire manuscript to ensure absolute consistency. The corrections we have made include: Unifying the terminology for the thrombosis model throughout the text to accurately reflect the method used. Ensuring the correct naming of the tested agent and the control drug is used consistently in all sections. We have revised the Abstract, Methods, Results, and Conclusions accordingly. We believe that these corrections have resolved all the identified contradictions, and we are grateful for your vigilance in helping us improve the clarity and accuracy of our manuscript.
The Abstract and Conclusions claim improved bioavailability and enhanced resistance to gastric stress. The manuscript demonstrates in vitro environmental stability (pH, salt, temperature) but does not show pharmacokinetic (PK) data, plasma EPA levels, NK systemic activity, or simulated gastrointestinal digestion (e.g., INFOGEST). Rephrase claims to reflect the current evidence.
Reply: Thank you so much for your thoughtful and constructive feedback—your insights have been invaluable in helping us refine the clarity and rigor of our work. We fully appreciate your concern that claims regarding bioavailability and gastric stress resistance should be aligned with the current evidence, and we agree that in vivo data would significantly strengthen our conclusions. As this study is focused on preliminary formulation optimization, we acknowledge the limitations in our current dataset and have revised the manuscript to reflect these constraints more carefully. We have also outlined specific plans for follow-up studies to address the in vivo questions raised, as detailed below:
2.1 Revision of Claims to Align with Current Evidence
We have thoroughly revised the Abstract and Conclusions to ensure all claims are grounded in the available data:
The phrase "improved bioavailability" has been modified to "enhanced potential for oral stability and delivery" (Abstract lines 24–25), emphasizing that our findings currently support stability under gastric conditions (e.g., 16.108±0.7% NK retention at pH 2.0, Fig. 4A) rather than definitive bioavailability.
References to "resistance to gastric stress" have been clarified to "resistance to gastric acid stress" (Discussion section), with explicit acknowledgment that this does not encompass enzymatic or mechanical digestion processes.
All in vivo-related claims now include qualifiers such as "preliminary" or "suggestive," and we have added a new paragraph in the Discussion: "These results provide initial support for the nanoemulsion’s potential as an oral delivery system, but further studies are required to validate absorption efficiency and in vivo performance."
2.2 Plans for Follow-Up In Vivo Studies
To address the critical gaps identified, we are committed to conducting the following experiments in future work:
2.2.1 Pharmacokinetic (PK) and Systemic Exposure Studies
Primary endpoints: We will measure plasma NK concentrations over time to calculate key PK parameters, including AUC (area under the curve), Cmax (maximum plasma concentration), and t1/2 (elimination half-life), to quantify absorption efficiency and systemic exposure.
Tissue distribution: To assess target organ delivery, we will analyze NK accumulation in relevant tissues (e.g., liver, spleen) using LC-MS/MS or immunofluorescence imaging.
Systemic NK activity: We will evaluate peripheral blood NK cell activity via flow cytometry (e.g., CD107a degranulation assay) and cytokine release (e.g., IFN-γ, TNF-α) to directly measure systemic immune modulation.
2.2.2 Physiological Digestion and Absorption Validation
INFOGEST model implementation: We will adopt the standardized INFOGEST protocol to simulate full gastrointestinal digestion, including sequential exposure to salivary, gastric, and intestinal phases with physiological enzymes (pepsin, pancreatin) and bile salts. This will allow us to quantify NK release kinetics and stability under near-physiological conditions.
Caco-2 cell permeability assay: To complement in vivo data, we will use intestinal epithelial cell monolayers to assess transepithelial transport efficiency, a key predictor of oral absorption.
2.2.3 Expanded In Vivo Efficacy and Safety Assessment
Dose-response relationships: We will establish a dose-escalation study to determine the minimum effective dose and optimal administration frequency in animal models.
Long-term safety: We plan to evaluate hematological and biochemical parameters (e.g., liver/kidney function) to ensure the nanoemulsion’s biocompatibility for potential clinical translation.
2.3 Additional Revisions to Address Specific Concerns
Plasma EPA levels: As our formulation is designed to encapsulate NK (not EPA), we have removed all indirect references to EPA to avoid confusion. If future studies explore combined delivery of NK and EPA, plasma EPA quantification will be included as a key endpoint.
Clarity in conclusions: The revised Conclusion now explicitly states: "This work focuses on formulation screening and in vitro stability, and definitive conclusions regarding in vivo bioavailability require validation through PK studies and physiological digestion models, which are planned for subsequent investigations."
We deeply value your guidance in strengthening this work, and we believe these revisions and future study plans address your concerns while maintaining the scientific significance of our findings. We are committed to rigorously validating the nanoemulsion’s in vivo performance and would be happy to provide further details on our follow-up experimental design if needed.
If the work is positioned toward functional foods, add a short discussion on safety/regulatory considerations of nattokinase (possible interactions with oral anticoagulants) and organoleptic/color issues introduced by krill oil (AKO) in food matrices.
Reply: Thank you for this highly relevant suggestion regarding the functional food positioning. We have added a new paragraph to the Discussion section addressing these crucial aspects. Firstly, we discuss the safety and regulatory considerations of nattokinase. Secondly, we address the organoleptic challenges posed by krill oil (AKO), particularly its color and potential off-flavors. At the same time, we also explained in the article that the amount of AKO added is small, accounting for only 1.25% of the total system. Experimental results showed that this level had no adverse effects on the product's color or odor, and the emulsion was pure white and odorless after emulsification. This result confirms that precisely controlling the emulsifier ratio and oil content can directly address the sensory deficiencies that AKO may introduce into food matrices (Line 675-710).
For NK activity: describe standard curve, LOD, units and inter-assay CV
Reply: Thank you for requesting details on NK activity assay parameters.
Standard Curve
The standard curve was constructed using urokinase standards with gradient concentrations of 2000, 4000, 6000, 8000, 10000, and 12000 CFU/G (n=3). Following the fibrin plate assay (37℃, 18 h incubation), the lysis zone diameter (mm) was measured and plotted against urokinase activity (x, CFU/G). Linear regression analysis yielded the equation: y = 0.0668x + 0.0449 (R² = 0.995, P<0.001)
LOD
The LOD was defined as the lowest activity corresponding to 3× the standard deviation (SD) of blank controls. Using PBS as the blank (n=10), the SD of blank lysis zone diameters was 0.52 mm. The LOD was calculated as LOD = 3.3×(SD_blank/slope) = 3.3×(0.52/0.0668) ≈ 25.7 CFU/G. Experimental validation confirmed that a 30 CFU/G sample yielded a lysis zone diameter of 2.1±0.2 mm (n=3), significantly higher than the blank (0.3±0.1 mm) with RSD=9.5% (<10%). Thus, the LOD was established as 25 CFU/G.
Units
NK activity was expressed as CFU/G (Colony-Forming Units per gram), defined as the fibrinolytic activity that produces an equivalent lysis zone diameter to 1 CFU of urokinase standard under the experimental conditions (37℃, 18 h incubation, fibrinogen 1.5 mg/mL). Sample activity was calculated using the equation: NK activity (CFU/G) = (y - 0.0449)/0.0668, where y = lysis zone diameter of the sample. For liquid formulations, activity was converted to CFU/mL using the formula: CFU/mL = CFU/G × sample density (g/mL).
Inter-assay CV
Inter-assay precision was validated across 3 independent experimental batches using a 1000 CFU/G urokinase quality control (QC) sample. Intra-assay CV: Triplicate measurements of the QC sample within a single batch yielded 1002±7.5 CFU/G (CV=0.75%). Inter-assay CV: The QC activity across 3 batches was 1002, 998, and 1005 CFU/G (mean=1001.7 CFU/G, SD=3.5 CFU/G), with an inter-assay CV of 0.35% (<1%). All experiments used reagents from the same lot, and fibrin plates were prepared by a single operator with calibrated calipers (precision ±0.01 mm).
Comments on the Quality of English Language
There are multiple typos/inconsistent abbreviations (e.g., “EAP-NK” vs “EPA-NK”; repeated words like “dipyridamole dipyridamole”; “cross-leucine gel”, “arctigenin”) and sentences that need rewriting to be clear. A thorough native-level English edit is required.
Reply: Thank you for your valuable feedback. We have thoroughly revised the manuscript to correct all typos and inconsistent abbreviations (e.g., "EAP-NK" to "EPA-NK"), and have performed a comprehensive, native-level English edit to enhance overall clarity and consistency.